# Pre-Pressure Optimization for Ultrasonic Motors Based on Multi-Sensor Fusion

**DOI:** 10.3390/s20072096

**Published:** 2020-04-08

**Authors:** Ning Chen, Jieji Zheng, Dapeng Fan

**Affiliations:** National University of Defense Technology, Deya Road No. 109, Kaifu District, Changsha 410073, China; 1024451173@126.com (J.Z.); fdp@nudt.edu.cn (D.F.)

**Keywords:** traveling wave ultrasonic motor, pre-pressure, contact state, power dissipation, optimization criterion

## Abstract

This paper investigates the pre-pressure’s influence on the key performance of a traveling wave ultrasonic motor (TRUM) using simulations and experimental tests. An analytical model accompanied with power dissipation is built, and an electric cylinder is first adopted in regulating the pre-pressure rapidly, flexibly and accurately. Both results provide several new features for exploring the function of pre-pressure. It turns out that the proportion of driving zone within the contact region declines as the pre-pressure increases, while a lower power dissipation and slower temperature rise can be achieved when the driving zones and the braking zones are in balance. Moreover, the shrinking speed fluctuations with the increasing pre-pressures are verified by the periodic-varying axial pressure. Finally, stalling torque, maximum efficiency, temperature rise and speed variance are all integrated to form a novel optimization criterion, which achieves a slower temperature rise and lower stationary error between 260 and 320 N. The practical speed control errors demonstrate that the proportion of residual error declines from 2.88% to 0.75% when the pre-pressure is changed from 150 to 300 N, which serves as one of the pieces of evidence of the criterion’s effectiveness.

## 1. Introduction

Ultrasonic motors, which work based on the converse piezoelectric effect and friction drive, tend to be the choice for precise actuators in diverse areas such as aerospace robots, manufacturing facilities, and especially for biomedical devices. Ultrasonic motors show their prominent advantages in biomedical devices like cell manipulation actuators [1], ear surgical devices [2], magnetic resonance imaging systems [3] and magnetic-compatible haptic interfaces [4,5]. This is due to ultrasonic motors’ features, which include excellent properties like high torque at low speed, high torque to weight ratio and no electromagnetic noise [6,7,8]. Pre-pressure is a non-negligible factor in all types of ultrasonic motors, for example traveling wave ultrasonic motors (TRUMs) [9], linear ultrasonic motors [10], 2-DOF ultrasonic motors [11], hybrid DOF ultrasonic motors [12] and so on. Among them, the studies of TRUMs are most representative. In TRUMs, piezoelectric ceramics are actuated by two-phase alternating voltages, the elliptical trajectories of stator particles form the traveling wave impelling the rotor revolve. Pre-pressure is applied from the rotor against the stator to assure the interface contact and friction drive. 

In processor studies, the performance sensitivities to the pre-pressure are mainly analyzed from the following three aspects. The first is the frequency characteristics. The first-order resonance frequency was investigated by the distributed numerical model and finite element model by Pirrotta [13]. His work found that the first-order resonance frequency was not sensitive to the preload force until the pre-pressure reaches a considerable value. However, these results were not verified by any further experimental results. Afterwards, both the resonant frequency and anti-resonant frequency under diverse pre-pressures were tested in Oh’s experiments [9]. He proved that the resonant frequency and anti-resonant frequency have a positive correlation with the preload force no matter whether the pressure is larger or smaller. Li [14] discovered that both frequencies do not increase monotonically with the pre-pressure. However, his data were collected from an impedance analyzer which only affords small-amplitude voltages (0–10 V). The second item is the contact properties. The contact state is the crucial contribution of pre-pressure for producing friction force and output torque. On this topic, Chen [15,16] developed a contact model via a semi-analytical model considering the radial slippage. The simulation results interpreted that the contact region and the driving zone become wider simultaneously when the pre-pressure increases, and the radial sliding inside the obstruction zone is also intensified. This useful yet incomplete analysis lacks any investigation on the contact state under load conditions. A contact model, which involves the distorted wave shape and standard stiffness of the contact layer, is built in [17]. This work also proves that the contact region becomes wider and the pressure at each contact point is raised with the increasing pre-pressure. Last but not least are the mechanical characteristics. Indeed, the mechanical characteristics integrate the results from the impedance characteristics and contact properties because the former determine the input power, and the latter affect the output power and torque. The most significant achievement in this aspect corresponds to Bullo [18]. He drew several three-dimension diagrams including speed, efficiency, output power both from simulations and experimental tests. However, the stator vibration amplitude was prescribed rather than adjusted by the differential functions, and the amplitude value is connected with voltage amplitude, driving frequency, applied load, and the initial pressure. His results also indicate that the incremental pre-pressure leads to a decreasing rotor speed under empty load conditions, but results in an increase of stalling torque.

Besides the above three performances concerning the preload force, this paper also proposes two new perspectives from the application viewpoint. One is the temperature rise arising from the power dissipation. As is well-known, TRUMs generate two energy conversion links, one is from electric energy to vibration energy based on the converse piezoelectric effect [19], and the other is from the vibration energy to the rotational energy via the friction between the stator and the rotor [20]. The energy losses occurring in the two processes not only reduce energy utilization but also significantly increase the internal temperature, which deteriorates TRUMs’ performances. Previous authors have made several contributions [19,21] to compensate for the temperature rise to obtain a better speed control performance. However, few researchers have paid attention to the role of pre-pressure in the temperature variation. The other is the speed stability. It is known that there exist speed fluctuations due to the discontinuities in the contact distribution on stator teeth and manufacturing errors. Similarly, the role of pre-pressure in velocity stability regulation has not been explored. Besides, the sources and associated factors of fluctuation need to be better quantified. The sensitive relationships between pre-pressure and motor performances spread all over the motor, which brings difficulties in deriving a uniform optimization criterion, so a relatively effective candidate scheme for the optimization region should be assured through the joint analysis of multiple performance examples from sufficient experimental tests. Hence, accurate and digital-controllable pre-pressure adjustment devices are needed. Nonetheless, changing of the preload forces is mostly accomplished by manual feeding through a screw [14,22], which lacks guidance accuracy and regulatory flexibility. Even worse, the motor shell has been modified in some preload-adjusting apparatus [14,23], which adds the extra workloads. 

In all, evaluation and optimization are the two main targets in this paper. In the evaluation process, modeling and experiment tests are synthetically implemented to evaluate the above performances under variable pre-pressure conditions. A hybrid model with the near-interface temperature rise is adopted to illustrate the frequency characteristics, contact properties, and temperature rise. A novel preload-control experimental apparatus is constructed from electric cylinder components, and a thin-film sensor is embedded inside a TRUM. In terms of the optimization process, the criterion considers the principles covering the lower temperature rise, the smaller speed stability, and the moderate mechanical properties. Finally, the optimization region of the preload force in a prototype motor is determined.

This paper is organized as follows: Section 2 focuses on the simulation model combined with power dissipation. Section 3 introduces the test bench with an electric cylinder and the thin-film temperature sensor for measuring the near-interface temperature. Section 4 analyzes the sensitivities of the main performance features in function of the preload forces from simulation and experimental tests. Section 5 proposes an optimization criterion according to the integrated results. Finally, conclusions and outlook are given in Section 6.

## 2. A Simulation Model with Power Dissipation

Preload force, which is imposed between motor stator and friction material, impels the motor to move. It is closely related to the performances of all motor components. Therefore a comprehensive model is required to account for the internal mechanisms in detail. In this work a TRUM60A ultrasonic motor (Chunsheng Ultrasonic Motor Co, Ltd, Nanjing, China) was investigated. Its stator operates in the B_09_ mode. In other words, the number (N) of traveling waves is equal to 9.

### 2.1. The Electromechanical Model

A TRUM is a typical electromechanical coupling system combining piezoelectric actuation with friction drive [24]. As shown in Figure 1a, two-phase piezoelectric ceramics are bonded symmetrically below the ring-shaped stator made of phosphor bronze material. When two-phase AC voltages are applied to the piezoelectric ceramics, the traveling wave impels the stator particles to vibrate with elliptic trajectories. Finally, the movements and accumulated energy are transferred to the rotor through a friction interaction for driving the terminate load. The friction layer of TRUM60A is polytetrafluoroethylene (PTFE), which has excellent time-varying stability [25]. What should be emphasized is that there is a gasket or disc spring between the bearing and the rotor. The component serves as the elastic element for withstanding the deformation caused by pre-pressure during assembly.

An electromechanical coupling TRUM model coalesces the models of both the driver and the motor itself. In this paper, a linear amplifying circuit is employed. After reproducing the actual circuit, the equivalent circuit model of piezoelectric ceramics is adopted. The ontology model originates from similar work by Kai [27], and the model equations and parameters are presented in Appendix A. Here, the critical parameters connected with the pre-pressure are mainly discussed and analyzed next. As shown in Figure 1b, due to the contact status and speed of the stator and the rotor, two feature points are generated on the stator teeth. One is the contact point used to distinguish whether the stator particles are embedded into the friction interface. The other is the sticking point, which is positioned at the point where the stator speed equals the rotor velocity. The stick points are applied to distinguish the driving zone and braking zone within the contact area. Usually, the contact length is defined as *X*c, and the length of the driving area is defined as *X*s. Since the TRUM60A has nine traveling waves, each wave occupies 40° of circumferential space along the whole circle of the stator ring. Therefore the above two parameters yield 0 ≤ *X*s ≤ *X*c ≤ 40°. Moreover, their values can be expressed as:(1)$$\begin{array}{l} Xc=\frac{20}{\pi }\arccos(\frac{h-z(t)}{R_{\mathrm{sc}}\xi })\\ Xs=\frac{20}{\pi }\arccos(\frac{\Omega_{r}R_{{}_{o}}^{2}}{\lambda hR_{\mathrm{sc}}\xi f_{s}}) \end{array}$$
where *ξ* represents the vibration amplitude, *h* denotes the half-thickness of the piezoelectric laminated board, *λ* represents the wavelength of the traveling wave, *f_s_* is the driving frequency and Ω_r_ is the angular velocity of the rotor. *R*(r) is defined as the transverse displacement distribution function along the radial direction. The value on the middle radius at the contact surface *R*_0_ is denoted as *R*_sc_. Different from the model discussed in [27], the modal amplitude is corrected by the method proposed by Li [14], which takes the preload effect into account. When defining *k* = *N*/*R*_o_, the contact parameters can be transformed into:(2)$$x_{0}=\frac{\pi Xc}{40k}\begin{array}{cc} {}_{}^{} & x_{1}=\frac{\pi Xs}{40k} \end{array}$$

Thus, the overlap between the stator and the rotor defines the compression of the springs along the circumferential direction, thereby the pressure distribution function of the contact interface can be given by:(3)$$p(x)=K_{f}R_{\mathrm{sc}}\xi [\cos(kx)-\cos(kx_{0})]$$
where *K_f_* represents the equivalent stiffness of the contact area. The output friction can be calculated by the surface integral of the forces within the contact area [25]. If the friction coefficient is defined as *μ*, *ε* means the width of the contact surface. Thus, the driving friction force can be read as:(4)$$\begin{array}{l} F_{R}=\mu \int_{R_{o}-\varepsilon /2}^{R_{o}+\varepsilon /2}\int_{-x_{0}}^{x_{0}}\mathrm{sgn}(\left|\Omega s(x)\right|-\left|\Omega r(x)\right|)p(x)dxdy\\ =\mu \int_{R_{o}-\varepsilon /2}^{R_{o}+\varepsilon /2}\int_{-x_{0}}^{-x_{1}}p(x)dxdy+\mu \int_{R_{o}-\varepsilon /2}^{R_{o}+\varepsilon /2}\int_{-x_{1}}^{x_{1}}p(x)dxdy+\mu \int_{R_{o}-\varepsilon /2}^{R_{o}+\varepsilon /2}\int_{x_{1}}^{x_{0}}p(x)dxdy\\ =2\mu K_{f}R_{\mathrm{sc}}\xi \varepsilon \left\{2[\frac{1}{k}\sin(kx_{1})-x_{1}\cos(kx_{0})]-[\frac{1}{k}\sin(kx_{0})-x_{0}\cos(kx_{0})]\right\} \end{array}$$

Obviously, the driving torque is derived from the multiplication between the average radius and the driving force. The summarizing result is depicted as:(5)$$T_{R}=NF_{R}R_{o}=2\mu \varepsilon K_{f}R_{\mathrm{sc}}R_{{}_{o}}^{2}\xi \left\{2[\sin(kx_{1})-kx_{1}\cos(kx_{0})]-[\sin(kx_{0})-kx_{0}\cos(kx_{0})]\right\}$$

Besides, the axial force, as well as the dynamic pressure on the stator surface, can be depicted as:(6)$$F_{z}=N\int_{R_{o}-\varepsilon /2}^{R_{o}+\varepsilon /2}\int_{-x_{0}}^{x_{0}}p(x)dxdy=2{\mathrm{NK}}_{f}R_{sc}\varepsilon \xi [\frac{1}{k}\sin(kx_{0})-x_{0}\cos(kx_{0})]$$

### 2.2. The Power Dissipation Model

The traveling wave motor operates with several energy conversion processes from the power source to the rotor rotation. Indeed, there are three main power dissipations which are the dielectric dissipation of the piezoelectric ceramics, the vibration dissipation of the stator, and the heat dissipation of the contact interface, respectively.

At first, the dielectric dissipation of the piezoelectric ceramics yields Equation (7). Here *ε_p_* means the dielectric constant of piezoelectric ceramic, tan*δ* denotes the dielectric loss coefficient, and *V*_piezo_ represents the volume of the piezoelectric wafer:(7)$$Q_{1}=2\pi f_{s}\varepsilon_{p}\frac{U_{m}^{2}}{h_{p}^{2}}V_{\mathrm{piezo}}\tan\delta$$

Secondly, quoting from the study of Lu [28], the mechanical damping dissipation can be displayed as: (8)$$Q_{2}=f_{n1}\int_{V_{piezo}}\delta \overset{-}{W_{sl}}dV=4\pi^{4}\xi^{2}N^{4}w(E_{s}I_{s}\eta_{s}+E_{p}I_{p}\eta_{p})/(\lambda^{3})$$
where *E_s_* and *E_p_* represent the equivalent elastic modulus of the stator and the piezoelectric strip, *I_s_* and *I_p_* denote their inertia moment, *η_s_* and *η_p_* are their mechanical damping coefficient, respectively. 

Last but not least is the power dissipation of the friction interface. The friction interface serves as the medium which accomplishes the transformation from the stator’s vibration to the rotor’s rotation. The friction drive gives rise to the radial friction losses and tangential ones between the stator and the rotor. The power losses stemming from the friction interaction can be determined by: (9)$$Q_{3}=N\frac{\lambda }{2\pi }\int_{kx_{o}}^{\pi -kx_{o}}\mu \Delta F(\nu_{s}-\nu_{r})(\nu_{s}-\nu_{r})d\theta =N\xi K_{f}\mu [M_{1}\nu_{sm}\nu_{r}+M_{2}(\nu_{sm}{}^{2})/2+M_{3}(\nu_{r}^{2})]$$
where *ν*_s_, *ν*_r_ are the linear velocity of the stator and the rotor, respectively, the *ν*_sm_ denotes the peak stator speed of the traveling wave, and the remaining coefficients can be depicted as:(10)$$\begin{array}{l} M_{1}=\pi -2kx_{o}+\sin(2kx_{o})\\ M_{2}=-0.33\cos(3kx_{o})+3\cos(kx_{o})-M_{1}\sin(kx_{o})\\ M_{3}=2\cos(3kx_{o})-\pi \sin(kx_{o})+2\varphi \sin(kx_{o}) \end{array}$$

Among the three power losses, it is verified that the power dissipation of the friction interface is the largest. Therefore the friction interface becomes the primary heat field which has been proved by Finite Element Analysis [28], which serves as a fundament for the placement of the temperature sensor discussed in Section 3. Finally, the shell temperature function, which derives from the thermodynamics transmission processes between the air and the motor, yields:(11)$$T=T_{\mathrm{air}}+\frac{Q_{1}+Q_{2}+Q_{3}}{\alpha S}(1-e^{-\frac{\alpha St}{C_{usm}}})$$
where *S* is the contact surface area, α represents the convective heat transfer coefficient, *C_usm_* is the heat capacity of the TRUM60A device. Moreover, it is worth noting that the near-interface temperature may be larger than *T* owing to the weak heat transfer capacity of the friction material. The rising temperature will lead to a speed decline and motor performance deterioration.

## 3. Experimental Setup

Figure 2a displays the mechanical test bench which is composed of an ultrasonic motor (TRUM60A), an incremental encoder (AFS60A, 65536 lines, SICK Corp., Waldkirch, Germany), an electric cylinder (EA0400, Huitong Corp., Nanjing, China), a torque sensor (9349A, Kistler, Sindelfingen, Germany) and a DC motor (55LYX04, YGGT Corp., Beijing, China) which is capable of generating any load profile in the current closed-loop mode. The electric cylinder is first employed to regulate the preload of the TRUM. As shown in Figure 2b, the axial force generated by the cylinder is transmitted from the piston rod to the motor shell. There is a force sensor (VC20A050, Vistle Corp., Guangzhou, China) which detects the real-time pressure. The pressure value can be displayed on a monitor or transmitted through a DA channel. Besides, the piston rod, the pressure sensor, and the pressboard were all limited inside a sleeve, which can assure the guidance accuracy of the axial movement. The design not only does not modify the TRUM’s structure but also assures the accurate and digital control of the pre-pressure. Moreover, an NTC film sensor (FWBM-337G104F, Fu Wen Corp., Shenzhen, China) is selected for satisfying the space restraint and response bandwidth of the temperature characteristics. The sensor measures the heating body through the polyimide whose thickness is only 15 µm, and its response time is only 0.1 s. As shown in Figure 2d, the sensor is placed against the stator’s teeth gap and near the interface where the heat is most concentrated. The signal is processed through a Wheatstone bridge which has been embedded into the circuit. 

Figure 2c introduces the framework of hardware and software. A driving & measurement circuit undertakes the function of amplifying the input signals. (*U*_A0_, *U*_B0_) and (*U*_A_, *U*_B_) are two pairs of voltages before and after amplifying, respectively. Meanwhile, the input voltages and currents are collected by voltage sensors and current sensors, respectively [29]. These channels are displayed from the signal group(*U*_aa_, *I*_aa_, *U*_bb_, *I*_bb_) to (*U*_a_, *U*_b_, *I*_a_, *I*_b_). The generation of (*U*_A0_, *U*_B0_) and the acquisition of (*U*_A_, *I*_A_, *U*_B,_
*I*_B_) are all conducted by a Field Programmable Gate Array (FPGA) control board (PXIE 7854R, NI Corp., Los Angeles, CA, USA). The FPGA programs can be transferred from the LABVIEW software, which provides abundant interfaces for algorithm generation and data acquisition [29]. The main parameters of the LABVIEW interface are all shown on the top of Figure 2c, where the left is the input parameters, and the output ones are listed in the right. In conclusion, a test system combining the flexible control of driving parameters and the accurate collection of the key parameters is built, which builds a foundation for exploring the sensitivities of motor performances under diverse pre-pressures.

## 4. Simulation and Experimental Results by Varying the Preload Force

The simulation model, as well as the measurement system, are used to comprehensively analyze the pre-pressure’s influences on several key performances including frequency characteristics, the stator/rotor contact states, the speed fluctuation, the temperature rise and the mechanical performance. Comparison and analysis are implemented for exploring the in-depth mechanism on how the preload force affects motion transfer and energy conversion.

### 4.1. The Stator/Rotor Contact

The stator/rotor contact is the premise that guarantees rotor rotation and torque output. If the driving parameters are unchanged, the contact status is mainly affected by the pre-pressure and the imposed load. Therefore, Figure 3a displays the contact length and the driving length for different cases of pre-stressing forces when the amplitude is 200 V and the frequency is 43 kHz. At first, the contact length generates a profound change when the preload force jumps away from zero value. Then the contact zone becomes more extensive with the slope 0.033°/N when the pre-pressure is increased from 100 to 570 N. Finally, the contact length increases with a higher slope until the contact area stretches over all the stator teeth. From the other curve, the length of the driving zone is equal to 40° when the stator is free from any pre-pressure. After that, the value gradually increases at a stable rate. However, it is always smaller than the contact length. This indicates that there exist driving areas and obstruction ones in the whole contact region. To be further, the proportion of the drive zone gradually decreases, which can also be verified from the proportion of the driving area in Figure 3b. These phenomena indicate that the stick-slip points gradually move toward the troughs of traveling waves, and the blocking effect is reinforced with the increasing pre-pressure, which induces more severe friction loss.

Once the motor operates with load, more output torque is needed to overcome the external torque, which causes the decline of rotor speed and the change of contact parameters. Figure 3c shows the contact parameters with respect to different loads from 0 to 0.9 N·m in function of pre-stressing forces. It is evident that the load has little effect on the contact length, which can also be verified by Equation (2). However, the driving length maintains a positive correlation with the load until the motor is blocked. In the blocking stage, the driving length returns to 40° again, which can be observed from the abrupt changes of the curves. In conclusion, if the preload is constant, the load torque does not affect the distribution of contact particles. And when the torque is constant, the contact area and the driving area gradually expand with the increase of the pre-pressures.

### 4.2. The Power Dissipation of the Motor

As discussed in Section 2, there are three main types of power dissipation inside the motor, and the temperature rise under different pre-pressures can be deduced. Figure 4 displays the respective time-variant energy loss in function of the preload forces when the amplitude and frequency are 200 V and 43 kHz, respectively. When the preload force is smaller than 460 N, the energy consumption increases as the preload force rises, which results from the gradually- widening contact area. By comparing the results of Figure 4a–c, it can be found that the dielectric loss maintains constant because the value is only related to the exciting voltage amplitude and the dielectric constant, which are both the intrinsic properties of piezoceramics. Furthermore, the stator damping loss and the friction loss follow similar positive-correlation laws until 460 N.

In order to exclude any additional factors, the starting temperature of every test should always be the same (28 °C), and the motor is turned off after the same time of operation (10 s). The results showing the motor speed and the near-interface temperature are displayed in Figure 5. The temperature rises immediately at the startup moment and drops rapidly once the driving voltages are withdrawn, which indicates that the accumulation and release of heat can be completed in a short time. Figure 5c demonstrates that the temperature rise achieves the highest value at 400 N, while the minimum temperature rise occurs when the preload is 300 N. With different temperature rise curves, the revolving speeds present different decline laws. It is evident that too fast speed decline is unfavorable for control and analysis.

### 4.3. The Speed Fluctuations 

The speed fluctuations can be attributed to the errors of the shaft system during manufacturing or assembly. Whether the pre-stressing force compensates or deteriorates, the speed fluctuations should be investigated. Figure 6 displays the real-time speed with the preload force varying from 150 to 500 N in steps of 20 N. The rotor speed is recorded when the frequency is 43 kHz, and the amplitudes are 200 and 240 V, respectively. To facilitate the quantitative evaluation of the fluctuation, the standard deviation is adopted, as shown in Figure 6. The results indicate that the motor speed fluctuation becomes smaller and smaller due to the increasing embedment degree between the stator and the friction layer. It can be attributed to sufficient contact with the larger preload force. However, when the preload reaches 475 N, the reinforced radial slipping deteriorates the velocity stability. Since the simulation state ignores the machining and assembly problems, such as uneven axis alignment and surface roughness in the stator and the rotor, it is difficult to simulate the fluctuations effectively. Furthermore, the dynamic pressure applied to the rotor may be the origin of speed fluctuations. The pressure also has an impact on the contact state. Here, the fluctuations are analyzed by the measurement of the speed and axial pressure in the subsequent experiments.

Based on the pressure *F*_z_ and the contact length *x*_0_ in Equation (1) polynomial fitting is implemented to derive the mapping relationship from the axial pressure to the contact length, as shown in Equation (12). When the pre-pressure is 150 and 200 N, Figure 7 displays the real-time speed, the angular position, the corresponding pressure and the calculated contact length, respectively. We can observe that the velocity fluctuation satisfies a stretch of the 360° cycle, which demonstrates that the fluctuation mostly comes from the un-centered assembly error of the rotor shaft. The wave cycle elongates with declining speed. When the preload is 150 N, the ratio of the maximum fluctuant zone (1.01°) inside the contact zone (24.15°) is 2.53%. If the pre-tightening force is 200 N, the proportion becomes 3.11% as the fluctuation range and the contact angle are respectively 0.9° and 28.94°. In conclusion, the speed fluctuation comes from both the pressure change and the axial assembly error. The change of contact parameters is also accompanied by a pressure change:(12)$$\begin{array}{l} X_{c}=2.7\times 10^{-16}F_{z}^{7}-6\times 10^{-13}F_{z}^{6}+5.4\times 10^{-10}F_{z}^{5}-2.5\times 10^{-7}F_{z}^{4}\\ +6.4\times 10^{-5}F_{z}^{3}-0.0088F_{z}^{2}+0.65F_{z}+0.72 \end{array}$$

### 4.4. The Mechanical Characteristics

With the aid of the electric cylinder and current sensors, a three-dimensional representation of the speed and the efficiency in function of the torque is shown in Figure 8 within the preload ranging from 150 to 400 N. The curves are obtained when the amplitude is 200 V and the frequency is 42 kHz. The dark blue areas are the stalling-torque areas, while the dark red ones are the maximum value zones of the respective curves. There exists a limit of the pre-stressing forces from which the motor performances fall. Figure 8b indicates the peak efficiency moves towards the larger-torque direction with the change of the preload forces. Deriving from the pictures, the blocking torque and maximum efficiency in function of the pre-pressures are displayed in Figure 8c. The blocking torque achieves the peak value in the moderate pre-pressure (260 N), so does the mechanical efficiency. The pre-pressure at the maximum efficiency point is 320 N. This is because the motor is close to the rotor-locked state when the pre-pressure is substantial, therefore the energy utilization is lower.

## 5. Discussion and Verification of the Optimal Preload Force

The above simulation and experimental results reveal how the performances are sensitive to the pre-stressing force. In this section, these key performances are reviewed, and an optimal criterion is proposed. 

### 5.1. Optimization Criterion

Some conclusions can be obtained from the above test results:(1)The velocity increases first and then decreases as the pre-pressure increases, and a low pre-pressure cannot provide sufficient friction force while a higher one causes more tangential friction zones;(2)When the pre-stressing force is lower, the velocity stability deteriorates because of the weakened constraints applied on the stator and the stability improves as the preload force gradually increases;(3)With the increase of the pre-pressure, the points both from the resonant frequency and the anti-resonant frequency gradually shift to the right due to the increasing stiffness;(4)The blocking torque achieves the peak value in the moderate pre-pressure, like the mechanical efficiency, however, the apexes are different from each other.

When limiting the pre-pressure range between 200 and 400 N, the key performances are summarized and arranged in Figure 9. Aiming at the lower open-loop speed stability, the preload force is supposed to be large enough for minimizing the dynamic fluctuations across the stator/rotor contact interface. Nonetheless, the moderate pressure is chosen based on the minimum temperature rise, which can decrease the undesirable changes in the tracking performances in the servo-control scheme. Based on the above analysis, the selected region is the yellow rectangle drawn in Figure 9. The region can not only meet the requirements of low-speed stability and small temperature rise but also the blocking torque and mechanical efficiency are in an ideal range. This method is different from others in that it gives priority to speed smoothness and temperature rise as reference points for pre-pressure assessments. 

### 5.2. Speed Control Performances

The validity of the optimization criterion is proved by the speed control performance, which not only stems from speed fluctuations but also reflects the effect of temperature rise. The speed closed-loop control with the target 72°/s is implemented under different preloads from 200 to 400 N. Considering the control target is located in the low-speed section, we select the voltage amplitude as the control parameter, the PID control method is adopted for every control process. Figure 10 displays the respective controlling results. It can be seen that residual control error decreases with the increasing preload forces when the preload force is less than 300 N and the error begins to rise once the pre-pressure exceeds 300 N. These results verify the effectiveness of pre-pressure optimization from one aspect, while other aspects will be further investigated.

## 6. Conclusions

Pre-pressure is one of the critical parameters that restrict the performance of an ultrasonic motor. The target of this paper is to comprehensively analyze the sensitivities of motor performances on the pre-pressure and to put forward a targeted optimization method. A simulation model with power dissipation and an integrated experimental facility with the preload adjustment device is adopted to analyze the laws from multiple perspectives. The preload adjustment is mainly designed by an electric cylinder and a pressure sensor, which is first employed in the preload change of the TRUM. Besides the stator/rotor contact state, the inspected properties cover rotor speed stability, near-interface temperature rise and mechanical properties. From these indicators, the speed stability, the temperature rise, the stalling torque, and the maximum efficiency are derived from the test results and drawn in a picture together. The operation makes these four indicators in function of the pre-pressure easy to distinguish in the candidate region according to different application targets. The optimization criterion in this paper contains three principles, which are the lower temperature rise, the smaller speed stability, and the moderate mechanical properties, respectively. Finally, the optimization region of the preload force is determined as [260, 320], and the smaller speed stability error in the optimization area verifies the criterion’s validity.

Besides, the contact state under different preloads is interpreted in simulation and experimental tests. This paper first proposes a polynomial formula for transferring the pressure into the contact angle, which helps us measure the interface contact properties indirectly. Therefore the speed fluctuations can be attributed to the contact variation. What should be mentioned that slight fluctuations of the compression preload force will occur during a TRUM’s physical life, especially in the application combining high speed with high load. The traditional preload component is a disc spring or thin-copper sheet, which cannot assure the correct imposing pressure in the total life-cycle. Luckily, the zero-stiffness structure similar to that of a PCB motor [30] can be used to ensure the pre-pressure, which will be the future optimization direction for all types of ultrasonic motors.

## Figures and Tables

**Figure 1 sensors-20-02096-f001:**
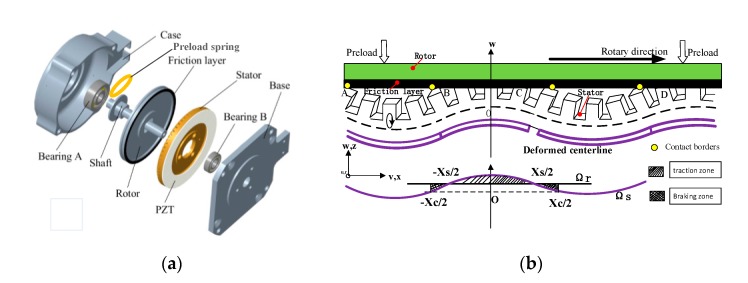
The mechanism of TRUM: (**a**) the motor structure [26]; (**b**) the traveling wave and contact state.

**Figure 2 sensors-20-02096-f002:**
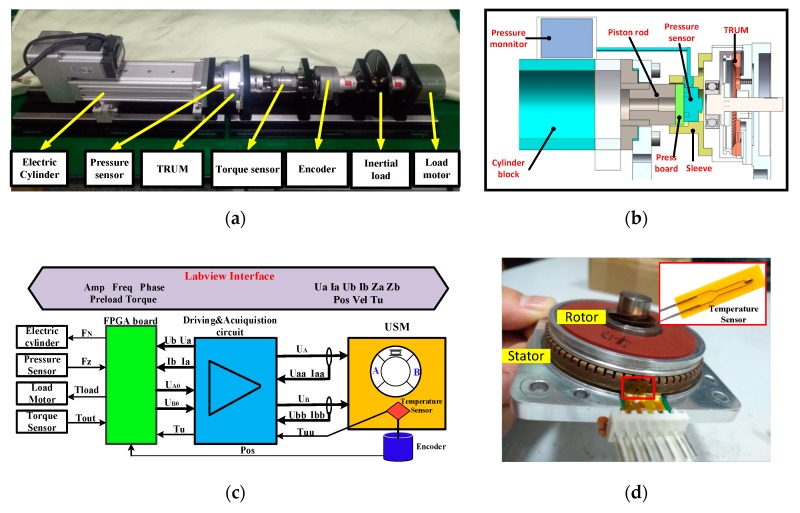
The integrated test system: (**a**) mechanical test bench; (**b**) structure scheme of the preload force regulating device; (**c**) framework and signal flow of the system; (**d**) the thin-temperature sensor placed near the interface.

**Figure 3 sensors-20-02096-f003:**
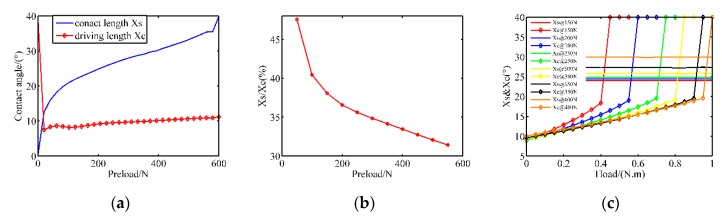
The contact parameters under diverse pre-pressures (simulation): (**a**) the contact angle and driving angel in no-load condition; (**b**) the proportion of driving zone in no-load condition; (**c**) the contact angle and driving angel with external load.

**Figure 4 sensors-20-02096-f004:**
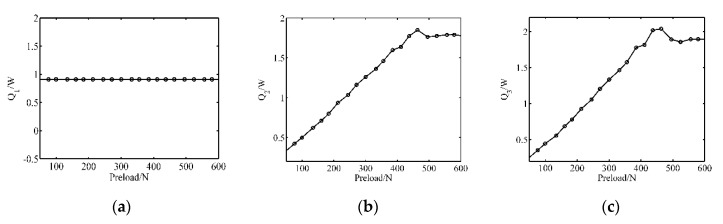
The respective power dissipation for the preload forces(Simulation): (**a**) Dielectric loss; (**b**) stator damping loss; (**c**) friction loss.

**Figure 5 sensors-20-02096-f005:**
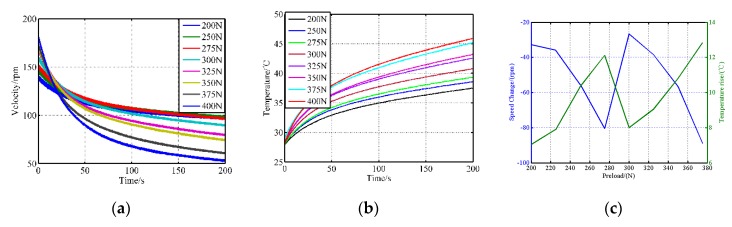
The startup-shutdown response under different pre-stressing forces (experimental): (**a**) the velocity response; (**b**) the temperature change; (**c**) temperature rise and speed decline.

**Figure 6 sensors-20-02096-f006:**
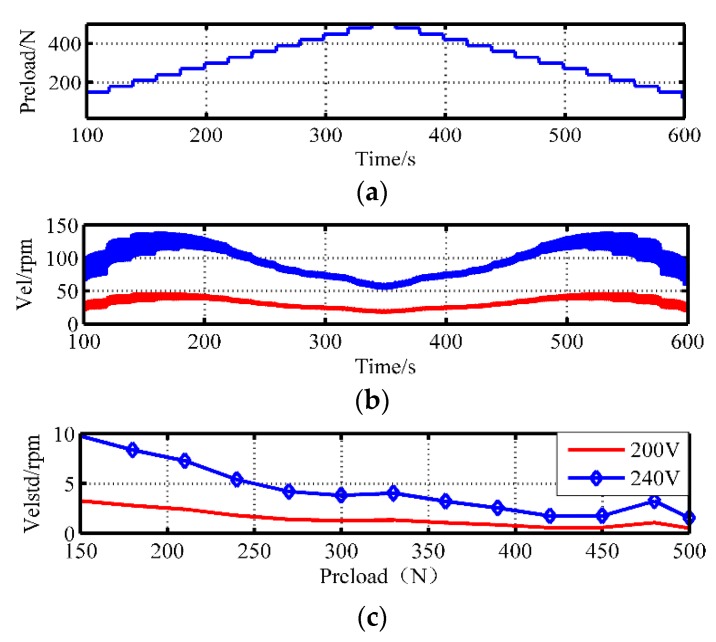
The speed stability under different preload forces: (**a**) staircase preloads; (**b**) output speed; (**c**) speed variance.

**Figure 7 sensors-20-02096-f007:**
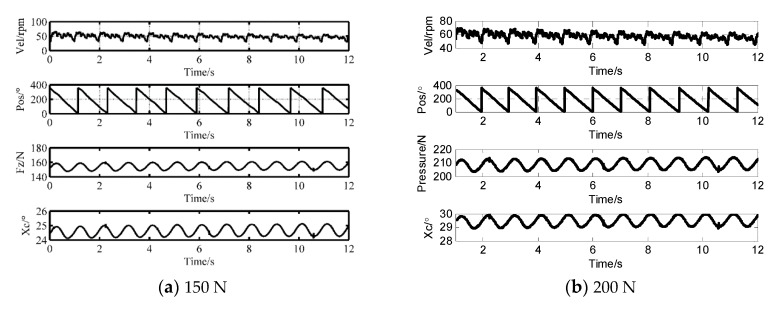
Time-domain analysis of dynamic pressure fluctuations and calculated contact angle.

**Figure 8 sensors-20-02096-f008:**
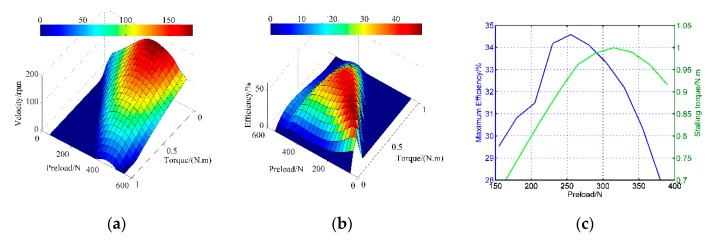
Simulation results of mechanical characteristics for different cases of pre-stressing forces: (**a**) revolving speed versus torque; (**b**) efficiency versus torque; (**c**) the calculated maximum efficiency and stalling torque.

**Figure 9 sensors-20-02096-f009:**
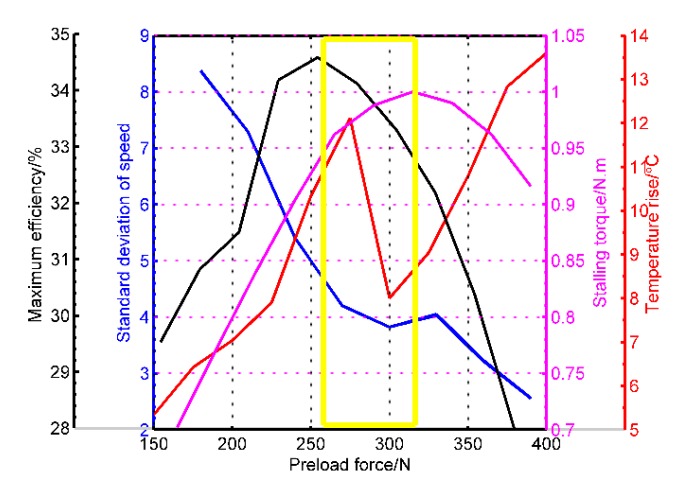
The diagram which interprets the optimization rule of the preload force.

**Figure 10 sensors-20-02096-f010:**
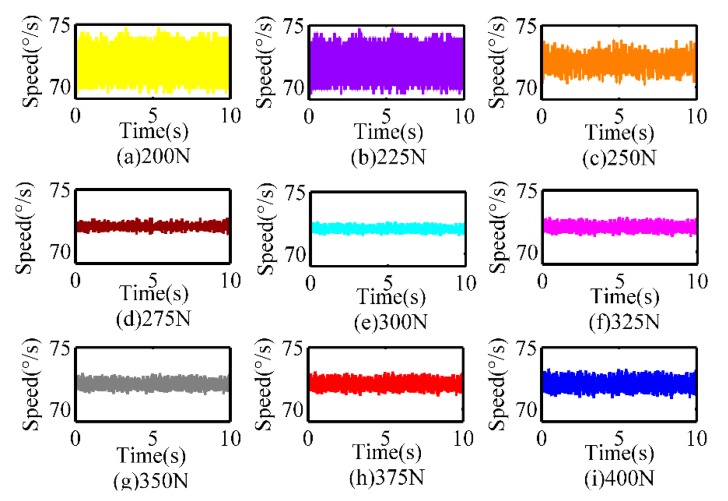
Comparison of the speed control performance under different pre-pressures.

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
