# Peer review of "Pre-Pressure Optimization for Ultrasonic Motors Based on Multi-Sensor Fusion"

_sensors, 2020, doi:10.3390/s20072096_

Round 1

Reviewer 1 Report

The main work of this paper is to build a test platform by taking the factors of pre-pressure and temperature into consideration when analyzing the influencing factors of motor performance, so as to provide references for the indexes that may be concerned (low temperature, speed stability and mechanical characteristics).In general, there have been lots of literatures on pre-pressure and thermal field analysis of USM. It would be better to do some more work on the effect of pre-pressure on interface loss.

1. The velocity component in formula (9) seems to be inconsistent with the following explanation;

2. Considering the distribution characteristics (such as material consumption and two interfaces), the temperature sensor in this paper is not necessarily accurate.

Author Response

My respected reviewer! Thank you very much for your valuable comments. According to your suggestions, the author checks all the variables and pictures in the paper. Please see the revised version of this article for details.Here are some responses for your comments:

1, In formula (9), the symbols νs,νr have been corrected to match the text content.

2 The interface temperature in the manuscript has been replaced by ‘near-interface temperature’, which is more objective and correct.

3 The format and layout of all equations has been corrected for matching the requirements of journal’s template.

Reviewer 2 Report

A few references is added presenting the applications of ultrasonic motor. It is better that the importance of this motor is presented stronger in the introduction. Medical applications and the fact that this motor is considered as an MRI safe actuator for image-guided surgical robotics are a good points that can be added to introduction. For example, the author can study the following paper that is close to their work.   

Measuring the temperature increase of an ultrasonic motor in a 3-tesla magnetic resonance imaging system

Line 199, add space before 'As' and remove '.' after Figure2

I suggest that you format the figure names in the text as simple as 'Figure 1' or 'Figure 2' with space and no dot. 

Author Response

My respected reviewer! Thank you very much for your valuable comments. According to your suggestions, the author checks all the variables and pictures in the paper. Please see the revised version of this article for details. Here are some responses for your comments:

1,In the introduction, I introduce the biomedical equipment like cell manipulation actuators [1], ear surgical devices [2], magnetic resonance imaging system [3] and magnetic-compatible haptic interfaces [4-5],which represent the wide application of ultrasonic motor in this specific area.

2 All the “Figure.1” have been replaced with ‘Figure 1’ and the “Equation.” have been replaced with ‘Equation’

Reviewer 3 Report

In general, the paper is organized and prepared well. I do not have any significant remarks. The study provided sufficient analytical study, optimization and tests. 

I have few minor remarks:

In Figure 3 the graphics could be larger, are hard to read. In Fig 3a) is a possibility to change colors (both lines are red). 

In line 251 is no space after ".". 

Author Response

My respected reviewer! Thank you very much for your valuable comments. According to your suggestions, the author checks all the variables and pictures in the paper. Please see the revised version of this article for details. Here are some responses for your comments:

1, the size and line color of Figure (3) has been improved.

2 The space in line 251 is withdrawn.

3 The format and layout of all equations has been corrected for matching the requirements of journal’s template.

Reviewer 4 Report

The authors present na interesting paper about optimization of pre-pressure control based on multi-sensor fusion applied to ultrasonic motors. However they should improve some issues on paper, namely:

1.- English writing review is recommended as the are some typos: "ultrasoni"; should replace "converse" with "inverse" to be coherent; ...

2.- Frequently (almost on all pages), authors do not respect senstence punctuation, because they either do not start the new sentence with a capital letter or do not give a space after a period.

3.- Some figures are somewhat distant from the point at wich tehy are referred, which make it difficult to follow the authors´explanation.

4.- References: the reference [28] in the bibligraphy is not mentioned in the text. The reference [30] mentioned in the text does not exist in the bibliography list!!

5.- It also seems strange to initiate a sentence with [ref]. It would be preferable to mention the authors´name and then include the [ref]. The name of the author of [4] (pag 2) must be written in lower case, for the writing to be consistent

6.- When you said "Applix 1" you mean "Appendix A"?

7.- In equation (4) is there no confusion between x0 and x1?

8.- The test situation and its conditions should be further detailed. Also, in relation to the exposure that the authors make of the experience, itsresults and conclusions, is not always easy for the reader to follow. A greater detail and perhaps the inclusion of some tables with the values resulting from the experiences would help a better understanding.

Finally, congratulations on your work

Author Response

My respected reviewer! Thank you very much for your valuable comments. According to your suggestions, the author checks all the variables and pictures in the paper. Please see the revised version of this article for details. Here are some responses for your comments:

1, The Engish writing errors and grammar catachresis have been corrected .

2 The wrong senstence punctuation have been corrected and some long sentences are divided into short ones to improve the readability.

3 Some reviews are made to assure that the figures are strictly correspond to the text described.

4 the reference [28] in the bibliography are supplemented in the text. The reference [30] mentioned in the text has been replaced by [29].

5 the sentences with [ref] have been corrected by adding the respective author’s name, such as the refer [23].

6The "Applix 1" has been replaced by "Appendix A"

7indeed, the parameter x1 is within the range of x0, and the real derivation process is described in formula(4).

8 The content of experimental setup has been extend ,the conditions including the driving parametetrs and load has been supplemented in the paper.

Round 2

Reviewer 1 Report

The authors have made amendment of the paper according to the reviewers' comments. The conclusions of the paper are useful to the researchers of ultrasonic motor.

Author Response

  My respected reviewer

    Thank you  very much for your recognition of my work.

    In this round, I have fixed a few punctuation marks on the foundation of the last revision.

  Best wishes for you